# Characterization of Composite Film of Gelatin and Squid Pen Chitosan Obtained by High Hydrostatic Pressure

**DOI:** 10.3390/polym15071608

**Published:** 2023-03-23

**Authors:** Ya-Ling Huang, Da-Ming Wang

**Affiliations:** 1Department of Seafood Science, National Kaohsiung University of Science and Technology, No. 142, Hai-Chuan Road, Nan-Tzu District, Kaohsiung 81157, Taiwan; 2Department of Chemical Engineering, National Taiwan University, No. 1, Section 4, Roosevelt Road, Da’an District, Taipei 10617, Taiwan

**Keywords:** chitosan, high hydrostatic pressure, composite film, peroxide value

## Abstract

In the present study, gelatin-based films incorporating squid pen chitosan obtained by high hydrostatic pressure (HHP chitosan) at varying proportions were prepared and their properties were compared with films containing untreated chitosan. The resulting films were characterized by analyzing the physical, morphological, mechanical and barrier properties. The addition of different ratios of HHP chitosan to the gelatin-based film yielded significant improvements in mechanical and moisture barrier properties. The reason for this might be that HHP chitosan contributed to a regular and dense microstructure of the composite films due to forming a three-dimensional network structure in gelatin-based films with enhanced intermolecular interactions. The FTIR spectra showed no new chemical bond formed by incorporating HHP chitosan into gelatin-based film. The SEM micrographs showed that the gelatin-based film fabricated with three types of chitosan had a homogeneous surface morphology, indicating good compatibility of the materials. Compared to the gelatin-based films containing untreated chitosan, films containing HHP chitosan significantly delayed oxidative deterioration in oil during storage. Therefore, the chitosan obtained by HHP treatment could have a potential application in edible gelatin-based films as packaging materials.

## 1. Introduction

Edible films are prepared from edible biopolymers and food-grade additives. The most common sources of edible film are polysaccharides, protein, lipids, or a mixture of these. These films are biodegradable and possess different properties [1]. Polymer blending can improve the physical properties of individual pure polymers. Polysaccharide was obtained from the cell walls of plants (e.g., cellulose and pectin), cell walls of crustaceans, fungi, insects (e.g., chitosan), and microorganisms (e.g., xanthan gum and glucan). These polysaccharides have different physical and chemical properties depending on their composition and biological sources. If the compatibility of these blended components is not carefully considered, film performance may be reduced. The incorporation of polysaccharides into protein matrices extends the functional properties of these components. Among these biopolymers, chitosan and gelatin have attracted great attention due to their biocompatible properties [1,2].

Gelatin is well known for its very good film-forming properties, which are applicable to the formation of transparent and flexible films. Chitosan is a cationic polysaccharide formed by the partial deacetylation of chitin. This polysaccharide has been widely used owing to its film-forming ability and antimicrobial properties. Natural polysaccharides and proteins are considered as biodegradable materials for edible films. Since chitosan has a good affinity with gelatin, the two materials can be formulated into edible films. The charged carboxyl group (COO–) from gelatin amino acid interacts electrostatically with the protonated amino group of chitosan to form a stable polyelectrolyte complex. This polyelectrolyte can be used to fabricate a transparent and homogeneous film. The molecular interactions between chitosan and gelatin caused modifications at the microscopic level, leading to improvements in mechanical properties, thermal stability, and biological activity of the resultant composite film. Blended films containing chitosan and gelatin promote barrier properties for food application through a high cross-linking interaction, leading to the formation of a denser and more compact matrix [2,3]. 

Wang et al. [4] reported that the mechanical and physical properties of blended chitosan-gelatin film were superior to those of single polymer-based films. To satisfy environmental concerns, these composite films may provide an alternative to synthetic packaging. The barrier and mechanical properties of these composite films are affected by the molecular structures and functional properties of chitosan extracted from different sources and under different conditions [5]. Chitosan has negligible biological activity in its native form due to the entrapment of functional groups within the complex structure of macromolecular. Some extraction methods have been used to extract chitosan from marine origins, including physical methods, chemical methods, and enzymatic methods. However, the enzymes used are expensive and inefficient and they had strict reaction conditions; chemical modification causes high organic solvent consumption. In order to overcome drawbacks, physical methods have been explored. 

In recent years, high hydrostatic pressure (HHP) has emerged as a technology that can effectively reduce extraction time and increase the efficiency of the process. Moreover, the main advantage of this technology is its ability to change the volume of the molecules and modify the structures of the components, which is related to transformations in cell wall polymers. Application of HHP on polysaccharides (i.e., cellulose and chitosan) induced the film-forming ability [6]. Preparation of cellulose acetate films by HHP (200–400 MPa for 5–10 min) treatment improved the physical properties and water resistance of films [7]. Through this physical invention, the properties of resulting films can be modified. The mechanism between macromolecules and HHP is probably related to the chemical structure within films [8]. In our previous study [9], we reported that chitosan obtained with HHP had enhanced physicochemical properties, such as fat binding capacity, water binding capacity, and water solubility index. 

Although many studies have focused on the film-forming properties of chitosan-gelatin film, there is no available literature on the properties of edible films composed of gelatin blended with chitosan extracted from squid pen using HHP. The interactions among the polymers can be achieved by varying the ratios, sources, and extraction methods to achieve desirable properties. Considering the advantages of films formed by gelatin and chitosan separately, we expected that combining gelatin with chitosan obtained by HHP with the gelatin in various ratios would produce better composite films than would each individual material alone. In addition, the molecular interactions between gelatin and chitosan obtained by HHP affect the structures and physical properties of composite films. Therefore, the aim of this study was to prepare composite films based on gelatin and chitosan obtained by HHP with various ratios, so as to improve the performance of the composite film. The aim of present study was to characterize the mechanical, thermal, optical, and moisture barrier properties of the composite films. Moreover, the composite films were evaluated for the preservation effect on oil. This research will offer food industries an opportunity to utilize a new composite film as potential packaging materials for food preservation.

## 2. Materials and Methods

### 2.1. Materials

Commercial cuttlefish chitosan was purchased from Charming & Beauty Co., Taoyuan, Taiwan. Commercial gelatin was purchased from Gemfont Co., Taipei, Taiwan. All other chemicals were of analytical grade.

### 2.2. Production of Chitosan from Squid Pens

*Illex argentinus* squid pens were provided from Shin Ho Sing Ocean Enterprise Co., Ltd., Taipei, Taiwan. Chitosan was prepared from squid pens (*Illex argentinus*) prior to high hydrostatic pressure (HHP) according to our previous work [8] and named HHP chitosan. In addition, the squid pen chitosan obtained without HHP treatment was named untreated chitosan. Commercial chitosan derived from cuttlefish was used as the control.

### 2.3. Preparation of Composite Films

The chitosan sample was dissolved in 1% (*v*/*v*) acetic acid to obtain 1% (*w*/*v*) chitosan solution. Single gelatin film-forming solution (4%, *w*/*v*) was prepared by mixing 4 g gelatin in 100 mL distilled water and then heating the solution at 90 °C for 30 min. The chitosan/gelatin composite films were formulated with chitosan to gelatin ratios of 100: 0, 80:20, 60:40, 40:60, 20:80, and 0:100 by mixing chitosan/gelatin film-forming solution with glycerol (0.4 mL) as a plasticizer. The mixture was stirred at 60 °C for 20 min. After ultrasound for 10 min, the final film-forming solution was poured into polystyrene petri dishes with a diameter of 9 cm and dried at 50 °C for 24 h. Then the dried film was peeled off and stored at 25 °C and 50% relative humidity for 48 h. 

### 2.4. Film Thickness

Film thickness was measured with a digital micrometer (No. 7327, Mitutoyo Manufacturing Co., Ltd., Tokyo, Japan) with precision of 0.001 mm. Ten random measurements were performed for each film sample. 

### 2.5. Mechanical Properties

Tensile strength (TS) and elongation at break (EAB) were measured with texture analyzer instruments (CT3, Brookfield Engineering Laboratories, Inc., Middleboro, MA, USA). The film strip (20 × 50 mm) was mounted on a sample grip. The initial instance of the fixture was 30 mm and the stretching speed was set at 30 mm/min. The force and distance were recorded while the film was stretched until break. Each film sample was tested at least five times. The TS and EAB were calculated by the following equations:(1)$$\mathrm{TS}\;(\mathrm{MPa})=\frac{F}{LM}$$
where *F* (N) is the maximum tension when the film broke, *L* (mm) is the thickness of the film, and *M* (mm) is the width of the film.
(2)$$\mathrm{EAB}\;(\%)=\frac{L_{1}-L_{0}}{L_{0}}\times 100$$
where *L*_1_ is the length of the elongation during the film fracture (mm) and *L*_0_ is the initial length of the film (mm).

### 2.6. Fourier Transformation Infrared Spectroscopy (FTIR) Analysis

The spectra were measured in the wavenumber region of 4000 to 400 cm^−1^ using a Perkin Elmer Frontier™ FTIR spectrometer (Perkin Elmer, Waltham, MA, USA) coupled with Platinum Diamond attenuated total reflection (ATR) during 200 scans with 4 cm^−1^ resolution. 

### 2.7. Thermal Analysis

#### 2.7.1. Thermogravimetric Analysis (TGA)

TGA was used to determine the thermal stability of the film samples. TG analysis was performed using PerkinElmer 8000 instrument (Waltham, MA, USA). About 6 mg of film sample was sealed in a ceramic pan and heated from 35 °C to 600 °C in inert nitrogen at a flow of 60 mL/min and heating rate of 10 °C/min. The DTG curve was obtained from differential TGA curves. 

#### 2.7.2. Differential Scanning Calorimetry (DSC)

The DSC of films was measured by differential scanning calorimeter (DSC 2500, TA Instruments, New Castle, DE, USA). The film sample (5 mg) was heated from 30 to 200 °C at a rate of 10 °C/min under a nitrogen atmosphere (50 mL/min).

### 2.8. Solubility

The solubility of films was measured according to the method of Roshandel-Hesari et al. [10]. Film samples were cut into 2 × 2-cm squares, weighed, and immersed in distilled water at room temperature for 24 h. After centrifugation at 1000× *g* for 10 min, the remaining undissolved film was dried at 105 °C until a final constant weight was obtained. The film solubility was calculated by the following equation:(3)$$\mathrm{Solubility}\;(\%)=\frac{\mathrm{initial}\;\mathrm{film}\;\mathrm{weight}-\;\mathrm{final}\;\mathrm{dried}\;\mathrm{film}\;\mathrm{weight}}{\mathrm{initial}\;\mathrm{film}\;\mathrm{weight}}\times 100$$

### 2.9. Water Vapor Permeability (WVP)

The water vapor permeability of the films was measured by gravimetric analysis, according to Dai et al. [11], with slight modifications. A beaker with a diameter of 5 cm was filled with 3 g granular calcium chloride. The mouth of the beaker was covered with each film sample and stored in a desiccator containing saturated KCl solution. The beaker was weighed every 24 h for 7 days. The WVP was calculated by using the following equation:(4)$$\mathrm{WVP}\;\left(g\cdot \mathrm{mm}/\mathrm{KPa}.h.m^{2}\right)=\frac{\Delta m\times d}{A\times \Delta T\times \Delta p}$$
where Δm (g) is the weight gain by the beaker over time, d (mm) is the film thickness, t (s) is the time interval, A (cm^2^) is the utilization area of the film, and ΔP (kPa) is the water vapor pressure difference across the film (3.185 kPa at 25 °C).

### 2.10. Color and Transparency

The color of the film samples was measured using a colorimeter (NE4000, Nippon Denshoku Industries Co., Ltd., Tokyo, Japan). The barrier properties of the films against ultraviolet and visible light were measured in transmittance mode using a UV-Vis spectrophotometer (UV-1280, Shimadzu, Kyoto, Japan). A standard white plate, with *L_s_** = 93.47, *a_s_** = −0.83 and *b_s_** = 1.33, was used for calibration. The total color difference (ΔE) was calculated by following equation:(5)$$\Delta E*=\sqrt{{\left(L*-L_{s}*\right)}^{2}+{\left(a*-a_{s}*\right)}^{2}+{\left(b*-b_{s}*\right)}^{2}}$$
where *L**, *a**, and *b** are the color values of the film sample and *L_s_**, *a_s_**, and *b_s_** are those of the standard white plate. 

The absorbance was measured at 600 nm and opacity values of the film were calculated according to the following equation: (6)$$\mathrm{Opacity}=\frac{\mathrm{Absorbance}600}{X}$$
where Absorbance600 is the absorbance value at 600 nm and x is the thickness of the films (mm).

High opacity values indicate low transparency.

### 2.11. Scanning Electron Microscopy (SEM)

Film morphology analysis was performed with a SEM (JSM-6330F, JEOL, Tokyo, Japan) to observe the surface and cross-section morphology of the films. The film samples were cryo-fractured under liquid nitrogen, mounted on aluminum stubs, and coated with a golden sputtering coater. Images were obtained at an accelerating voltage of 10 kV and at varying degrees of magnification.

### 2.12. Peroxide Value (PV)

Erlenmeyer flasks (125 mL) were filled with 5 g of soybean oil, covered with different film samples, and sealed with rubber band. These flasks were stored at room temperature for 0, 7, 14, 21, 28, and 35 days. A portion (30 mL) of 3:2 (*v*/*v*) acetic acid and isooctane was poured into the above flask containing soybean oil. Saturated potassium iodine was added and the mixture was titrated with 0.1 N sodium thiosulfate using starch solution as an indicator until the violet color disappeared. A blank control sample, without film covering the flask, was prepared under identical conditions. All analyses were performed in triplicate. Peroxide values were expressed as milliequivalents of peroxide index per kilogram of oil.

### 2.13. Statistical Analysis

Data were analyzed by a one-way analysis of variance (ANOVA) in SPSS 16.0 (SPSS Inc., Chicago, IL, USA). The differences between means were analyzed by Duncan’s Multiple Range and *p* < 0.05 was considered to be statistically significant.

## 3. Results and Discussion

### 3.1. Thickness

Thickness is an important parameter that influences the utilization of films as food packaging. Moreover, the rate of water vapor permeation, tensile strength, and elongation of the film are dependent on the thickness. Table 1 shows that all edible films increased in thickness, regardless of type of chitosan added to the composite film. In general, the thickness of the composite film depended on the deacetylation degree of the chitosan [12]. Among the chitosan samples used in this work, the gelatin-based films containing HHP chitosan had the lowest thickness. This is possibly due to the changes in numbers of hydrogen bonding between the NH_3_ groups of the HHP chitosan backbone and the –OH groups of gelatin.

### 3.2. Mechanical Properties of Films

Edible films having a higher tensile strength (TS) value are considered to be able to protect food from mechanical force. Table 1 shows that the highest value was obtained at 100C:0G and the lowest value at 0C:100G, regardless of whether untreated or HHP chitosan was incorporated into the composite film. The results showed that a higher ratio of chitosan was obviously and significantly (*p* < 0.05) related to higher tensile strength. The film with a higher chitosan ratio had lower elongation at break. High negative correlations were found between the elongations and tensile strengths of the films incorporated with untreated chitosan (*r* =−0.92) and those with HHP chitosan (*r* = −0.93), respectively. A similar trend was found in the study of Prateepchanachai et al. [13] for composite films composed of gelatin and chitosan. Especially, the incorporation of HHP chitosan in the composite film significantly (*p* < 0.05) increased the TS value. This result indicated that the HHP treatment produced a significant increase in the TS of the composite films because of the changes caused by the considerable amount of chitosan, with a low molecular weight, interacting with gelatin in the composite films. Compared to the elongation at break (EAB) value of the pure gelatin film, the EAB of composite films increased with increasing content of chitosan regardless of chitosan sources. Among the composite films fabricated with three types of chitosan, the film containing HHP chitosan had the highest value. This implied that the gelatin-based film incorporated with HHP chitosan had a great reinforcement effect due to the increased rigidity, leading to a decrease in EAB. Therefore, the addition of squid pen chitosan obtained with HHP is expected to produce mechanically sturdier materials for edible film applications. Kim et al. [14] observed that the buckwheat and tapioca starch films treated by HHP (300–600 MPa for 20 min) had significantly higher TS and lower EAB compared to that of films treated by thermal processing methods (90 °C). Concerning the properties of packing films, the mechanical properties, such as TS and EAB, provide resistance to external stress during the packaging process. Compared with the pure gelatin film, the composite film had much better mechanical properties. Tensile strength is associated with the mechanical resistance of films due to the cohesion force between polymer chains [15]. Moreover, the rigidity and strength of the networks within the edible film were probably increased by electrostatic interaction between the positively charged chitosan and negatively charged gelatin [16]. In terms of EAB, higher rigidity of the edible film was accompanied by lower malleability. Therefore, the application of HHP to prepare chitosan influenced the mechanical properties of gelatin-based films. 

### 3.3. FTIR Analysis

FTIR analysis was used to study the differences in the functional groups in the edible films. The FTIR spectra of gelatin-based films containing different chitosan sources are shown in Figure 1. As for the pure gelatin film, O-H stretching and N-H stretching vibrations between gelatin chains appeared in the range of 3200–3500 cm^−1^. The spectra of the gelatin-based film spectra with additions of three different types of chitosan had characteristic peaks of O-H stretching that shifted higher toward 3288 cm^−1^. From the spectra of the composite film, it was found that the chitosan might have interacted with the gelatin molecules through hydrogen bonds between O-H and N-H groups. The peaks around 2958 cm^−1^ corresponded to asymmetric stretching vibrations of NH^3+^ and CH, representing the amide-B bond [17]. Moreover, the spectrum of pure gelatin film showed characteristic peaks at 1633, 1538, and 1238 cm^−1^, corresponding, to amide-I (C=O stretching vibration/coupling to CN stretching), amide-II (C–N stretching), and amide III (C–N stretching and N–H bending), respectively. Regarding the peaks of amide-I (CO) and amide-III (CN and NH), the spectra of the composite films presented significant shifts when the proportion of chitosan in the gelatin film was increased. The shift was attributed to the alteration of secondary structure gelatin polypeptide chains after the gradual addition of chitosan to the film [18]. The signal of the pure chitosan film displayed a group of characteristic saccharide bands located in the range of 1100–900 cm^−1^. All the composite films fabricated with chitosan, albeit from different sources, and gelatin had similar FTIR spectra, as demonstrated by those of the squid pen chitosan before and after HHP treatment [9]. The positions of amide-I and amide-II peaks in the films shifted to higher wavenumbers with increases in the amount of added chitosan, indicating that NH group gradually interacted with the chitosan during casting [19]. The intensity of the amide III bands of composite films weakened compared to that of pure gelatin film. This was attributed to the likely precipitation of carboxyl groups of gelatin in electrostatic interactions with oppositely charged –NH^+3^ groups of chitosan. This finding was in agreement with a study by Wang et al. [4], who reported gelatin crosslinking with chitosan nanoparticles. Overall, the FTIR results exhibited similar IR spectra for all the films incorporating chitosan obtained with/without HHP and gelatin at the same composition ratio, indicating no structural changes in the functional groups of the composite films. 

### 3.4. Thermogravimetric Analysis (TGA)

Thermogravimetric analysis is a method wherein the mass of a substance is monitored as a function of temperature in controlled atmospheric conditions. The thermal behavior of composite films can be observed from the DTG curves shown in Figure 2. The DTG curve of pure gelatin film had three maxima at 81.57 °C, 160.70 °C, and 267.91 °C, with corresponding weight losses of 9.47%, 6.82%, and 70.34%, respectively. Incorporation of untreated chitosan in the gelatin-based film at a ratio of 80:20 led to the formation of a thermally stable matrix, for higher degradation temperatures appeared at around 85.92 °C, 212.31 °C, and 317.03 °C and their weight loss of 20.53%, 65.39%, and 13.54%, respectively. The pure HHP chitosan and pure untreated chitosan films had similar initial weight losses. The mass loss in the first stage was related to the evaporation of water. The second stage weight loss of pure HHP chitosan occurred at 215.75 °C, which was similar to pure untreated chitosan. The thermal decomposition pattern of the film indicated that the HHP did not significantly affect the thermal stability of the gelatin-based films. However, Kumar et al. [20] reported degradation temperatures of 320 °C for chitosan. When the amount of the three types of chitosan in the composite film was increased, the maximum value of this DTG peak reached 323.44 °C. This can possibly be attributed to crosslinking with the NH_2_ group of chitosan in the composite film, which would increase the degradation temperatures [21]. Therefore, the composite mixtures are more thermal stable than pure chitosan films.

### 3.5. Differential Scanning Calorimetry (DSC)

As shown by the DSC analyses of chitosan, gelatin, and composite films in Figure 3, an endothermic peak was observed at 79.14 °C for gelatin. This peak is related to the glass transition temperature (Tg). However, endothermic peaks of its biocomposites were obtained as 90.18 °C, 140.99 °C, 141.41 °C, and 144.85 °C for untreated composite films with chitosan to gelatin ratios of 20:80, 40:60, 60:40, and 80:20, respectively. For pure untreated chitosan, the endothermic peak was at 211.21 °C, which is possibly related to the α-relaxation of chitosan chains [14]. This value is similar to the results obtained by Hiremani et al. [22], who observed that the Tg of chitosan film with added glycerol was 170 °C. Furthermore, the temperatures of the endothermic peaks of the composite films with various proportions of HHP chitosan ranged from 88.98 to 216.45 °C. It was speculated that the Tg of the films comprising chitosan was related to the increase in the number of hydroxyl groups forming intermolecular hydrogen bonds. The increase in the Tg can be attributed to the intermolecular interactions between chitosan and gelatin. The increased thermal stability of the composite film containing chitosan can be attributed to the formation of hydrogen bonds within the polymer network. However, no significant difference was found in the temperatures of the endothermic peaks between pure HHP chitosan and pure untreated chitosan in the films. A similar trend of improvement in thermal properties has been found in chitosan/gelatin nanofiber using nozzle-less electrospinning [23]. The three types of chitosan can be blended well with gelatin during process and improved the thermal stability of the composite film because the film-forming molecules formed many chemical bonds with water and glycerol molecules. The pure chitosan and its composite films did not exhibit a melting temperature, which implied that the most polysaccharide did not melt as a result of the amorphous state of these films. 

### 3.6. Solubility and Water Vapor Permeability (WVP)

The presence of water in food products significantly influences the rate of deteriorative reaction during storage. The solubilities and WVPs of films composed of chitosan prepared by different treatment methods and gelatin in different ratios are shown in Table 2. The pure gelatin film exhibited a higher solubility than those of films containing chitosan. This result was attributed to chemical nature of gelatin, which promoted interactions between surface adsorption sites and water molecules. A similar result was obtained in the study of Al-Maqtari et al. [24] for chitosan/gelatin films incorporating *Pulicaria jaubertii* extract. The composite films prepared with higher ratios of chitosan showed significantly (*p* < 0.05) lower solubility values. Increasing the amount of untreated chitosan in the gelatin-based film reduced the water solubility from 84.16 to 32.07%. When HHP chitosan was added, the solubility was significantly (*p* < 0.05) lower than those of composite films containing untreated chitosan. This result agreed with a previous report by Jridi et al. [25], which indicated that increasing the chitosan content in the composite film led to decreased solubility. The film with HHP chitosan had the lowest value (20.48–84.16%). This finding was attributed to HHP having eliminated the charges on the surface of the chitosan in the composite film. This difference can be ascribed to the degree of deacetylation of the chitosan used in the preparation of the edible film. The application of HHP on polyvinyl alcohol-chitosan composite film improved WVP [26]. Similarly, Molinaro et al. [27] indicated that HHP-treated gelatin had a reduction in WVP, which is attributed to the changes in the stability of hydrogen bonds. It was speculated that HHP treatment reduced the passage of water molecules through the polymeric network, resulting in improving the barrier property of the composite films. To maintain the freshness of food, a lower WVP value of edible film is preferred. The development of food packaging films is intended to reduce the exchange of water between the food products and the environment. Table 2 shows that the pure gelatin film had high WVP, possibly due to the presence of a high level of hydrophilic amino acids within the gelatin. The addition of chitosan to the gelatin-based film significantly (*p* < 0.05) reduced the WVP value, with that of the film with a high ratio of HHP chitosan film being the lowest. This finding is similar to one in a report by Yadav et al. [28]. In other words, the film fabricated with chitosan after HHP treatment exhibited a moisture barrier property. This observation was in accordance with Mohammadi et al. [29], who observed that increasing the addition of chitosan in a composite film improved the better water vapor barrier performance due to the increased interaction between the gelatin chain and chitosan. The edible film containing HHP chitosan, which had a low molecular weight, could easily absorb the water vapor and improve the diffusion step due to the presence of a hydrophilic domain within the film matrix [30].

### 3.7. Color and Transparency

The colors of the gelatin-based films containing the three types of chitosan are shown in Table 3. No matter which chitosan was used, the incorporation of a higher content of chitosan in the gelatin-based film caused a significant (*p* < 0.05) reduction of the L value. A similar result was reported by Prateepchanachai et al. [12] for chitosan-gelatin edible film processed by CO_2_ treatment. The film fabricated with HHP chitosan demonstrated a significantly (*p* < 0.05) lower a value but a higher b value and ΔE compared with those of the film prepared by adding untreated chitosan. These results indicated that the addition of HHP chitosan into gelatin-based film increased the yellowness coloration. The change in color parameters may have been due to the color of the chitosan itself, produced by HHP treatment. Regarding the opacity of films, the lowest opacity values indicated the highest transparency. The opacity values range from 0.24 to 1.16, with significant differences (*p* < 0.05) among the different films. The three types of chitosan could change the transparency of the composite film (Table 3). Increasing the content of chitosan in the gelatin-based film led to higher opacity values, whereas untreated chitosan made the film more opaque. The highest opacity value of the composite film composed of untreated chitosan and gelatin at a ratio of 80:20, respectively, indicated a less transparent film, followed by the film with HHP chitosan. It was speculated that this could be attributable to the formation of poly-anion/cation complexes. The differences in the transparency of the edible film were attributed to the light scattering caused by the chitosan of different molecular weights. When the film matrix contracted, the decreases in the polymer inter-chain spacing allowed less light to pass through the film [31]. The opacity values of the films are affected by the color, thickness, and components [29]. Films with high opacity could protect food from oxidizing reactions, which is a potential application as food packaging materials.

### 3.8. Scanning Electron Microscopy (SEM)

As shown in Figure 4, the surfaces of edible films fabricated with chitosan and gelatin in different ratios were analyzed by SEM. The surface morphology of the composite films was more even and smooth than that of the pure gelatin films, regardless of chitosan type. As increases in the chitosan content in the gelatin-based film, the composite films had very uniform and homogeneous structures without any apparent phase separation between chitosan and gelatin. With higher amounts of chitosan incorporated in the gelatin, a dense and uniform structure distributed throughout the polymer matrix was observed in the composite films. There was no apparent phase separation, representing the interaction and high compatibility between two polymers in the blend. This structure may have been due to the sufficient cross-linking between chitosan and gelatin; these characteristics were attributed to the compatibility of the three types of chitosan and gelatin. The results indicated that the chitosan and gelatin were mixed well at different ratios, as good compatibility among these components was evident within the film matrix. In addition, the incorporation of HHP chitosan with a higher degree of deacetylation in the film yielded a more homogenous surface structure. The possible reason was the number of hydrophilic functional OH and NH_2_ groups, which, in turn, increased the number and strength of the hydrogen bonds between polymers. Wang et al. [4] reported that rice protein hydrolysate edible/chitosan film fabricated with ultrasound treatment displayed a uniform and cohesive internal structure without cracks. It was speculated that physical treatment could be an effective method to improve the interaction between protein and chitosan components in the film matrix.

### 3.9. Peroxide Value (PV)

As shown in Table 4, the initial peroxide value (PV) of soybean oil, whether or not it was covered with the edible film, was approximately 5 meq/kg. The Codex Alimentarius proposes that the PV for refined oils such as soybean oil shall be below the limit of 10 meq/kg to be considered fresh [32]. When the oil was not covered with film, the PV increased as the storage duration increased. The maximum value (51.07 meq/kg) was observed after 35 days. In the case of pure gelatin film, it was found to have a higher peroxide value than those of other composite films. This result was related to the higher transparency of the gelatin film, which accelerated the oil oxidation process [33]. Increasing the addition of chitosan in the gelatin-based film decreased the PV, regardless of which type of chitosan was used, and the PVs of films with HHP chitosan were much lower than those of the others. On day 35, the lowest PV was achieved with the 80:20 film, and the PV was significantly (*p* < 0.05) lower in soybean oil covered with HHP chitosan films than that in oil covered with films containing commercial or untreated chitosan. The results indicated that gelatin-based films prepared with HHP chitosan provided better protection against lipid oxidation than did film made with untreated chitosan. Wang et al. [4] indicated the gelatin film incorporated with tea polyphenol and chitosan nanoparticle prepared by electrospray could delay the rancidity of oil during storage for 14 days (PV value of 30 meq/kg). However, the oxidation of soybean oil was effectively retarded over 14 days of storage periods when it was packaged with films containing HHP chitosan. Therefore, the addition of HHP chitosan in the gelatin-based film had a positive effect on retarding the lipid primary oxidation.

## 4. Conclusions

This study prepared a gelatin-based film incorporating chitosan obtained by HHP. The incorporation of HHP chitosan in the gelatin-based film improved the tensile strength, decreased the water vapor permeability, enhanced the thermal stability, and reduced the transparency. The FTIR spectra showed there is no change in the chemical structure of films containing commercial chitosan, untreated chitosan, and HHP chitosan. The SEM micrographs of films fabricated with three types of chitosan displayed a compact and smooth morphology, indicating high compatibility between the chitosan and gelatin. Notably, storage tests with soybean oil showed lower peroxide values for oil covered with films containing HHP chitosan than for oil covered with films containing untreated chitosan and commercial chitosan. The film containing HHP chitosan significantly delayed oil oxidation during storage. Therefore, this film can be used as an alternative to synthetic materials and has great potential for edible food packaging applications. Further studies are needed to study the antimicrobial activity and application of the film composed of HHP chitosan and gelatin towards extending the shelf life of food. 

## Figures and Tables

**Figure 1 polymers-15-01608-f001:**
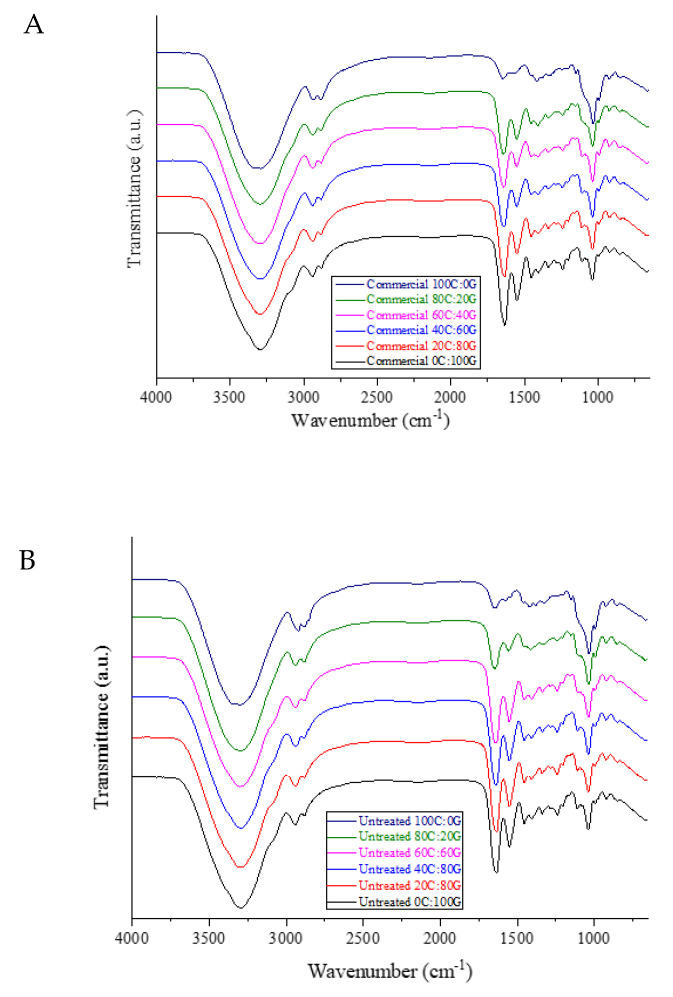
FTIR spectra of gelatin-based film incorporated with commercial chitosan (**A**), untreated chitosan (**B**), and HHP chitosan (**C**).

**Figure 2 polymers-15-01608-f002:**
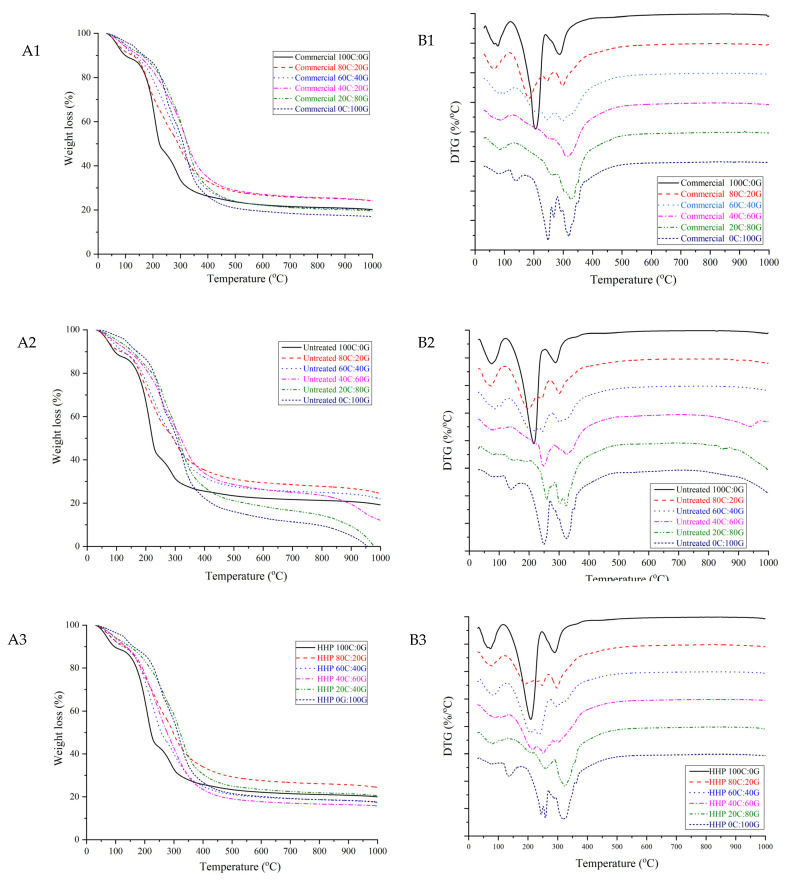
TGA curves of gelatin-based film incorporated with commercial chitosan (**A1**), untreated chitosan (**A2**), and HHP chitosan (**A3**); DTG curves of gelatin-based film incorporated with commercial chitosan (**B1**), untreated chitosan (**B2**), and HHP chitosan (**B3**).

**Figure 3 polymers-15-01608-f003:**
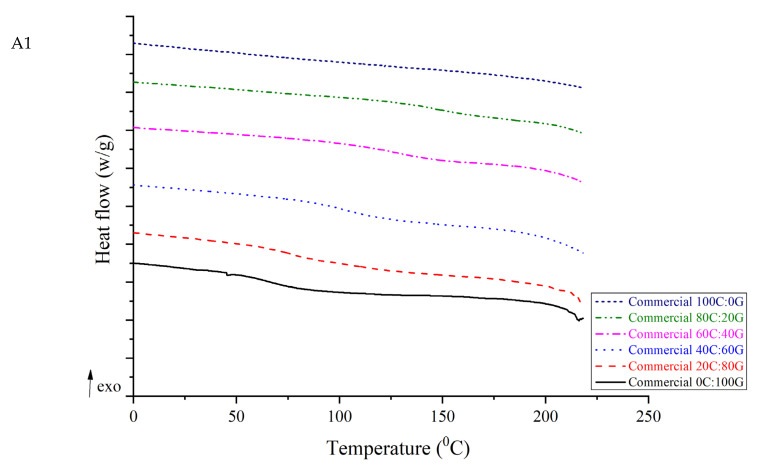
DSC curves of gelatin-based film incorporated with commercial chitosan (**A1**), untreated chitosan (**A2**), and HHP chitosan (**A3**).

**Figure 4 polymers-15-01608-f004:**
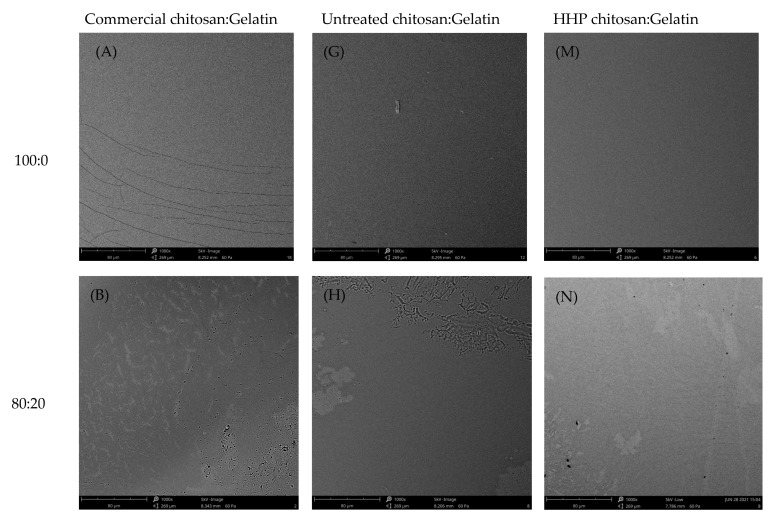
SEM images at 1000× magnification of surface of the composite films containing commercial chitosan (**A**–**F**), untreated chitosan (**G**–**L**), or HHP chitosan (**M**–**R**) and gelatin in different ratios.

**Table 1 polymers-15-01608-t001:** Mechanical properties of composite films.

Film Samples	Thickness (mm)	TS (MPa)	EAB (%)
Commercial			
100C:0G	0.081 ± 0.004 ^f^	7.54 ± 0.11 ^d^	26.41 ± 0.86 ^h^
80C:20G	0.095 ± 0.004 ^e^	7.52 ± 0.25 ^d^	29.68 ± 0.27 ^g^
60C:40G	0.106 ± 0.101 ^d^	6.39 ± 0.42 ^e^	31.91 ± 0.72 ^f^
40C:60G	0.121 ± 0.004 ^c^	5.31 ± 0.67 ^f^	33.06 ± 0.42 ^f^
20C:80G	0.134 ± 0.005 ^b^	5.29 ± 0.49 ^e^	39.29 ± 0.63 ^e^
0C:100G	0.168 ± 0.002 ^a^	4.21 ± 0.32 ^g^	60.88 ± 0.28 ^a^
Untreated			
100C:0G	0.074 ± 0.003 ^g^	9.24 ± 0.21 ^b^	26.87 ± 1.27 ^h^
80C:20G	0.083 ± 0.002 ^f^	7.84 ± 0.20 ^d^	33.14 ± 0.86 ^f^
60C:40G	0.104 ± 0.002 ^d^	6.42 ± 0.27 ^e^	35.17 ± 1.28 ^f^
40C:60G	0.112 ± 0.002 ^d^	6.33 ± 0.15 ^e^	43.15 ± 1.02 ^d^
20C:80G	0.131 ± 0.001 ^b^	6.03 ± 0.33 ^e^	50.26 ± 1.16 ^c^
0C:100G	0.168 ± 0.002 ^a^	4.21 ± 0.32 ^g^	60.88 ± 0.28 ^a^
HHP			
100C:0G	0.066 ± 0.002 ^h^	11.08 ± 0.18 ^a^	27.18 ± 1.20 ^h^
80C:20G	0.075 ± 0.002 ^g^	9.55 ± 0.29 ^b^	38.97 ± 0.04 ^e^
60C:40G	0.102 ± 0.001 ^d^	8.91 ± 0.19 ^c^	44.62 ± 0.67 ^d^
40C:60G	0.107 ± 0.002 ^d^	8.37 ± 0.54 ^c^	53.80 ± 1.42 ^b^
20C:80G	0.124 ± 0.002 ^c^	7.40 ± 0.62 ^d^	58.33 ± 1.24 ^a^
0C:100G	0.168 ± 0.002 ^a^	4.21 ± 0.32 ^g^	60.88 ± 0.28 ^a^

Values are means ± standard deviations. ^a–h^ Different superscripts within the same column indicate significant differences between treatments (*p* < 0.05).

**Table 2 polymers-15-01608-t002:** Solubility and WVP of composite films.

Film Samples	Solubility (%)	WVP (g.mm)/(m^2^.h.kPa)
Commercial		
100C:0G	33.67 ± 0.20 ^h^	0.172 ± 0.005 ^d^
80C:20G	35.16 ± 0.31 ^g^	0.175 ± 0.012 ^d^
60C:40G	35.90 ± 0.73 ^g^	0.180 ± 0.004 ^d^
40C:60G	42.47 ± 0.51 ^e^	0.185 ± 0.005 ^c^
20C:80G	45.85 ± 0.16 ^d^	0.189 ± 0.003 ^b^
0C:100G	84.16 ± 0.19 ^a^	0.195 ± 0.001 ^a^
Untreated		
100C:0G	32.07 ± 0.19 ^h^	0.161 ± 0.006 ^e^
80C:20G	43.60 ± 0.85 ^e^	0.167 ± 0.003 ^e^
60C:40G	47.35 ± 1.26 ^c^	0.177 ± 0.004 ^d^
40C:60G	48.26 ± 1.16 ^c^	0.180 ± 0.005 ^c^
20C:80G	54.74 ± 1.34 ^b^	0.187 ± 0.008 ^b^
0C:100G	84.16 ± 0.19 ^a^	0.195 ± 0.001 ^a^
HHP		
100C:0G	28.48 ± 0.04 ^i^	0.155 ± 0.001 ^f^
80C:20G	38.44 ± 1.43 ^f^	0.157 ± 0.009 ^f^
60C:40G	43.44 ± 1.43 ^e^	0.167 ± 0.013 ^e^
40C:60G	43.96 ± 0.86 ^e^	0.175 ± 0.009 ^d^
20C:80G	48.82 ± 0.78 ^c^	0.180 ± 0.005 ^c^
0C:100G	84.16 ± 0.19 ^a^	0.195 ± 0.001 ^a^

Values are means ± standard deviations. ^a–i^ Different superscripts within the same column indicate significant differences between treatments (*p* < 0.05).

**Table 3 polymers-15-01608-t003:** Color and opacity of composite films.

Film Samples	*L**	*a**	*b**	ΔE	Opacity
Commercial					
100C:0G	95.15 ± 0.06 ^b^	−0.52 ± 0.02 ^h^	3.80 ± 0.01 ^d^	6.14 ± 0.04 ^e^	0.68 ± 0.11 ^e^
80C:20G	95.37 ± 0.08 ^b^	−0.40 ± 0.01 ^g^	4.07 ± 0.01 ^c^	6.17 ± 0.06 ^e^	0.69 ± 0.01 ^e^
60C:40G	95.69 ± 0.01 ^b^	−0.40 ± 0.01 ^g^	5.49 ± 0.01 ^b^	6.99 ± 0.01 ^c^	0.62 ± 0.05 ^e^
40C:60G	95.88 ± 0.01 ^b^	−0.41 ± 0.02 ^g^	6.21 ± 0.01 ^a^	7.46 ± 0.01 ^b^	0.50 ± 0.01 ^f^
20C:80G	96.18 ± 0.01 ^a^	−0.36 ± 0.02 ^f^	6.10 ± 0.01 ^a^	6.94 ± 0.07 ^c^	0.37 ± 0.02 ^g^
0C:100G	96.50 ± 0.02 ^a^	−0.10 ± 0.01 ^b^	0.86 ± 0.01 ^l^	3.57 ± 0.01 ^c^	0.24 ± 0.05 ^h^
Untreated					
100C:0G	88.73 ± 0.24 ^e^	0.08 ± 0.01 ^a^	2.71 ± 0.02 ^g^	11.26 ± 0.04 ^a^	1.16 ± 0.01 ^a^
80C:20G	94.77 ± 0.01 ^c^	−0.41 ± 0.02 ^g^	2.48 ± 0.01 ^h^	5.80 ± 0.01 ^g^	1.53 ± 0.01 ^c^
60C:40G	95.62 ± 0.01 ^b^	−0.32 ± 0.01 ^e^	2.05 ± 0.01 ^i^	4.85 ± 0.01 ^i^	1.25 ± 0.05 ^d^
40C:60G	95.82 ± 0.39 ^b^	−0.39 ± 0.01 ^g^	2.64 ± 0.01 ^g^	4.60 ± 0.01 ^j^	0.63 ± 0.02 ^e^
20C:80G	95.26 ± 0.02 ^b^	−0.17 ± 0.01 ^c^	1.33 ± 0.01 ^j^	3.94 ± 0.10 ^l^	0.34 ± 0.01 ^g^
0C:100G	96.50 ± 0.02 ^a^	−0.10 ± 0.02 ^b^	0.86 ± 0.01 ^l^	3.57 ± 0.01 ^n^	0.24 ± 0.05 ^h^
HHP					
100C:0G	93.76 ± 0.01 ^d^	−0.28 ± 0.01 ^e^	2.88 ± 0.01 ^f^	6.87 ± 0.01 ^d^	0.71 ± 0.01 ^e^
80C:20G	94.80 ± 0.01 ^c^	−0.41 ± 0.03 ^g^	2.90 ± 0.01 ^f^	5.96 ± 0.01 ^f^	0.97 ± 0.01 ^b^
60C:40G	95.54 ± 0.06 ^b^	−0.33 ± 0.05 ^e^	3.44 ± 0.01 ^e^	5.63 ± 0.05 ^h^	0.51 ± 0.01 ^f^
40C:60G	95.83 ± 0.01 ^b^	−0.21 ± 0.02 ^c^	2.63 ± 0.01 ^g^	4.93 ± 0.02 ^i^	0.42 ± 0.01 ^g^
20C:80G	96.43 ± 0.02 ^a^	−0.26 ± 0.01 ^d^	1.81 ± 0.01 ^i^	4.01 ± 0.02 ^k^	0.40 ± 0.01 ^g^
0C:100G	96.50 ± 0.02 ^a^	−0.10 ± 0.02 ^b^	0.86 ± 0.01 ^k^	3.57 ± 0.01 ^m^	0.24 ± 0.05 ^h^

Values are means ± standard deviations. ^a–n^ Different superscripts within the same column indicate significant differences between treatments (*p* < 0.05).

**Table 4 polymers-15-01608-t004:** PV value of composite films.

Film Samples	PV (meq/Kg)					
	0 Day	7 Days	14 Days	21 Days	28 Days	35 Days
Without film	5.08 ± 0.01 ^a^	11.11 ± 0.01 ^a^	19.57 ± 0.02 ^a^	22.26 ± 0.01 ^a^	47.70 ± 0.02 ^a^	51.07 ± 0.29 ^a^
Commercial						
100C:0G	5.06 ± 0.02 ^a^	7.95 ± 0.01 ^d^	10.37 ± 0.10 ^d^	15.89 ± 0.01 ^b^	38.13 ± 0.03 ^c^	44.29 ± 0.18 ^c^
80C:20G	5.07 ± 0.02 ^a^	6.36 ± 0.01 ^e^	9.05 ± 0.02 ^e^	13.35 ± 0.01 ^i^	25.44 ± 0.01 ^g^	41.02 ± 0.03 ^f^
60C:40G	5.07 ± 0.02 ^a^	9.53 ± 0.01 ^c^	9.01 ± 0.05 ^e^	14.62 ± 0.01 ^e^	28.60 ± 0.02 ^f^	42.26 ± 0.02 ^e^
40C:60G	5.07 ± 0.02 ^a^	9.54 ± 0.01 ^c^	10.55 ± 0.01 ^c^	14.94 ± 0.01 ^d^	31.77 ± 0.01 ^e^	43.22 ± 0.08 ^d^
20C:80G	5.06 ± 0.02 ^a^	9.54 ± 0.01 ^c^	10.56 ± 0.01 ^c^	15.25 ± 0.01 ^c^	38.16 ± 0.02 ^c^	44.50 ± 0.03 ^c^
0C:100G	5.07 ± 0.02 ^a^	11.13 ± 0.01 ^a^	11.39 ± 0.08 ^b^	15.90 ± 0.01 ^b^	41.58 ± 0.01 ^b^	47.99 ± 0.04 ^b^
Untreated						
100C:0G	5.07 ± 0.02 ^a^	9.54 ± 0.01 ^c^	10.53 ± 0.02 ^c^	14.30 ± 0.02 ^f^	34.97 ± 0.03 ^d^	42.29 ± 0.03 ^e^
80C:20G	5.07 ± 0.02 ^a^	7.94 ± 0.01 ^d^	8.92 ± 0.08 ^f^	13.68 ± 0.01 ^h^	25.43 ± 0.05 ^g^	38.16 ± 0.02 ^i^
60C:40G	5.07 ± 0.02 ^a^	11.12 ± 0.01 ^a^	10.39 ± 0.03 ^d^	13.99 ± 0.01 ^g^	28.60 ± 0.01 ^f^	39.08 ± 0.08 ^h^
40C:60G	5.07 ± 0.02 ^a^	11.13 ± 0.01 ^a^	10.42 ± 0.04 ^c^	14.30 ± 0.01 ^f^	34.96 ± 0.01 ^d^	40.70 ± 0.02 ^g^
20C:80G	5.07 ± 0.02 ^a^	11.12 ± 0.01 ^a^	10.50 ± 0.01 ^c^	14.62 ± 0.01 ^e^	38.14 ± 0.01 ^c^	43.86 ± 0.04 ^d^
0C:100G	5.07 ± 0.02 ^a^	11.13 ± 0.01 ^a^	11.39 ± 0.08 ^b^	15.90 ± 0.01 ^b^	41.58 ± 0.02 ^b^	47.99 ± 0.04 ^b^
HHP						
100C:0G	5.07 ± 0.02 ^a^	7.95 ± 0.01 ^d^	9.05 ± 0.01 ^e^	13.34 ± 0.01 ^i^	31.80 ± 0.01 ^e^	39.72 ± 0.02 ^h^
80C:20G	5.07 ± 0.02 ^a^	6.36 ± 0.01 ^e^	7.53 ± 0.01 ^h^	11.76 ± 0.01 ^l^	22.25 ± 0.01 ^g^	34.98 ± 0.01 ^k^
60C:40G	5.07 ± 0.02 ^a^	9.53 ± 0.01 ^c^	8.44 ± 0.02 ^g^	12.08 ± 0.01 ^k^	25.44 ± 0.01 ^g^	36.56 ± 0.03 ^j^
40C:60G	5.07 ± 0.02 ^a^	9.54 ± 0.01 ^c^	9.04 ± 0.01 ^e^	12.72 ± 0.02 ^j^	28.59 ± 0.01 ^f^	38.14 ± 0.01 ^i^
20C:80G	5.07 ± 0.02 ^a^	10.33 ± 0.01 ^b^	10.56 ± 0.02 ^c^	13.34 ± 0.01 ^i^	31.79 ± 0.01 ^e^	39.74 ± 0.08 ^h^
0C:100G	5.07 ± 0.02 ^a^	11.13 ± 0.01 ^a^	11.39 ± 0.08 ^b^	15.90 ± 0.01 ^b^	41.58 ± 0.50 ^b^	47.99 ± 0.04 ^b^

Values are means ± standard deviations.^a–l^ Different superscripts within the same column indicate significant differences between treatments (*p* < 0.05).

## Data Availability

The data presented in this study are available on request from the corresponding author.

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
