# Peer review of "Characterization of Composite Film of Gelatin and Squid Pen Chitosan Obtained by High Hydrostatic Pressure"

_polymers, 2023, doi:10.3390/polym15071608_

Round 1

Reviewer 1 Report (Previous Reviewer 3)

Introduction is short the authors should evidence the advantages of gelatin edible films comparing with similar product based on polysaccharides, or why they chose these biopolymers

Introduction part: please rephrase this sentence: chemical modification causes low organic solvent consumption. Compared with other modification methods.

In section Result and discussion figure 1A2 the explanation are not correct please correct de percentage.

Figure 1A3 please replace Intensity with transmission.

Figure 2 B3 is for untreated chitosan not for HHP chitosan

Author Response

Dear reviewer#1,

Thank you for your thoughtful review. Our revised manuscript entitled Characterization of composite film of gelatin and squid pen chitosan obtained by high hydrostatic pressure” (polymers-2274288) has been carefully revised and improved according to the reviewer's comments. Any changes made in the revised manuscript have been highlighted in red "Please see the attachment." in the box.

Reviewer 2 Report (Previous Reviewer 1)

Authors have sufficiently made the changes suggested.

Author Response

Dear reviewer#2,

Thank you for your thoughtful review.

Reviewer 3 Report (Previous Reviewer 2)

Reviewer #2 read the response and the revised manuscript.

The authors have kindly answered to the comments of Reviewer #2 and revised the manuscript.

The authors have re-evaluated the experimental data and have addressed the concerns of the reviewer #2.

Therefore, Reviewer 2 recommend to publish this revised manuscript to “Polymers”.

Author Response

Dear reviewer#3,

Thank you for your thoughtful review.

This manuscript is a resubmission of an earlier submission. The following is a list of the peer review reports and author responses from that submission.

Round 1

Reviewer 1 Report

1. Title must be rephrased as "Characterization of composite film of gelatin and squid pen chitosan obtained by high hydrostatic pressure".

2. line 15: remove word "due to"

3. line 15: a more compact microstructure of what?

4. line 15: "The thermal stability" should be "Thermal stability"

5. line 16: "a chitosan to gelatin ratio of 50:50" should be "50:50 chitosan to gelatin ratio"

6. line 21-22: "a chitosan-to-gelatin ratio of 80:20" should be "80:20 chitosan-to-gelatin ratio"

7. line 82: What does "1% (w/v) acetic acid" mean?

8. All equations must be numbered.

9. line 114: The statement "wavelength region of 4,000 to 600 cm−1" must be corrected as cm-1 is not wavelength rather wavenumber.

10. line 117: heading "2.7. Thermogravimetric analysis" must be "2.7. Thermal analysis", as DSC is not gravimetric analysis rather it gives calorific values. Moreover line 117-125 must be redrafted to separate data related to both TGA and DSC, for better understanding of readers.

11. line 129: what do authors mean by "at 1,004g" ?

12. line 138: remove word "glass"

13. line 143: "of the cup" must be "by the beaker" 

14. line 165: What do authors mean by "High transparency values indicated low transparency of the film sample"?

15. line 168: scanning electron microscope must be abbreviated as SEM not JSM.

16. Table 1: Different superscripts used in table 1 must be explained in detail. 

17. Table 1: data needs to be checked carefully. Samples Commercial (0C:100G), Untreated (0C:100G) and HHP (0C:100G) are same thing, then why different values? If this difference in values is being attributed to instrumental or operator error, how authors would defend that data for other samples is correct?

18. line 206, 219: no need to use full form of already abbreviated phrase. 

19. Figure 1: Authors mentioned in section 2.6 that FTIR analysis has been carried out in range of 4000-600 cm-1, but figure 1 contains data from 4000-400cm-1. Why? Moreover, FTIR of pure chitosan for all three samples (Commercial C100, Untreated C100 and HHP C100) appear to be similar, then how authors claim that their effect would be different in composites? A strong justification is required here. 

20. Y-axis values of Figures B1, B2 and B3 must be removed.

21. In Table 1 authors claimed HHP chitosan with compact film (less film thickness and high TS) in comparison to untreated chitosan, but in "section 3.8 TGA" authors reported high degradation of HHP chitosan than untreated chitosan. It looks confusing, because highly crosslinked materials degrade less.

22. DSC graphs required. Authors must add DSC curves of all samples in section 3.5.

23. Table 2: data needs to be checked carefully. Samples Commercial (0C:100G), Untreated (0C:100G) and HHP (0C:100G) are same thing, then why different values? If this difference in values is being attributed to instrumental or operator error, then how authors would defend that data for other samples is correct?  

24. Table 3: data needs to be checked carefully. Samples Commercial (0C:100G), Untreated (0C:100G) and HHP (0C:100G) are same thing, then why different values? If this difference in values is being attributed to instrumental or operator error, then how authors would defend that data for other samples is correct?  

25. Table 4: data needs to be checked carefully. Samples Commercial (0C:100G), Untreated (0C:100G) and HHP (0C:100G) are same thing, then why different values? If this difference in values is being attributed to instrumental or operator error, then how authors would defend that data for other samples is correct?  

26. Cross sectional SEM images are not very clear. 

Reviewer 2 Report

The reviewer carefully read the submitted manuscript entitled “Characterization of composite film of squid pen chitosan obtained by high hydrostatic pressure and gelatin”.  The authors have prepared the gelatin-based films incorporating squid pen chitosan obtained by high hydrostatic pressure (HHP chitosan) at varying proportions, and have evaluated the effect of the incorporation of different proportions of chitosan on the physical attributes and mechanical properties of the films, and have compared them with the gelatin-based films incorporating commercial chitosan and untreated one. 

The authors have insisted that the addition of different ratios of HHP chitosan to the gelatin-based film yielded significant improvements in mechanical and moisture barrier properties. The mechanical properties in Table 2 shows that tensile strength (TS) of the films with HHP chitosan is higher than the other films. However the values of elongation at break (EAB) cannot be compared since the EAB values for 0C:100G for each sample are not equal to each other.  

In section 3.5 (line 316), the authors have mentioned that HHP treatment could produce a chitosan with a smaller molecular weight and improve the thermal stability of the composite film.  Table 2 shows that the glass transition temperatures (Tg) of films depend on the ratio of chitosan-to-gelatin.  However the reviewer does not think that Table 2 shows any differences between the films with commercial, with untreated and with HHP chitosan.  Moreover the reviewer cannot understand the reason why a chitosan with a smaller molecular weight improve the thermal stability of the composite film.

Preoxide values (PV) in Table 5 show that the PV value for 80C:20G for HHP chitosan is lowest.  This result seems to show that the film with a chitosan to gelatin ratio of 80:20 has the best performance as mentioned in this manuscript.  However the other experimental results in this manuscript do not support this result.

  The scanning electron microscopy (SEM) image shows that the composite films containing HHP chitosan and gelatin at a ratio of 20:80 have clearly rougher surface patterns.  The authors do not explain why this ratio shows such a structure so clearly.

For the above reasons, the reviewer cannot recommend the publication of the manuscript to the journal “Polymers”.

Reviewer 3 Report

Interesting results and novelty work. A paper focused on Characterization of composite film of squid pen chitosan obtained by high hydrostatic pressure and gelatin. I recommend the publication after revision.

In the abstract, the problem that the authors are trying to solve is not clearly specified, and the methods of characterizing the films obtained are not specified. 

The Introduction section is too short, insufficiently documented, it does not highlight the advantages of using this technique compared with others techniques presented in specialized literature.

Row 137 CaCl2 is written incorrectly

Please add the cities for apparatus used for the instrumental analyses (FTIR, TG, UV-Vis, SEM).

In section Results please correct row 305 (chitosan HHP not HPP).

It should be more  discussions regarding the results obtained by authors compare with others similar researches, highlighting the advantages of using the proposed technique.